# Are the classic false belief tasks cursed? Young children are just as likely as older children to pass a false belief task when they are not required to overcome the curse of knowledge

Siba Ghrear[1]*, Adam Baimel[2], Taeh Haddock[1], Susan A. J. Birch[1]

**1** Department of Psychology, University of British Columbia, Vancouver, British Columbia, Canada,
**2** Department of Psychology, Health and Professional Development, Oxford Brookes University, Oxford, United Kingdom

* siba.ghrear@psych.ubc.ca

**Data Availability Statement:** All relevant data are uploaded within the paper and its Supporting information files.

## Abstract

The question of when children understand that others have minds that can represent or misrepresent reality (i.e., possess a 'Theory of Mind') is hotly debated. This understanding plays a fundamental role in social interaction (e.g., interpreting human behavior, communicating, empathizing). Most research on this topic has relied on false belief tasks such as the 'Sally-Anne Task', because researchers have argued that it is the strongest litmus test examining one's understanding that the mind can misrepresent reality. Unfortunately, in addition to a variety of other cognitive demands this widely used measure also unnecessarily involves overcoming a bias that is especially pronounced in young children—the 'curse of knowledge' (the tendency to be biased by one's knowledge when considering less-informed perspectives). Three- to 6-year-old's (n = 230) false belief reasoning was examined across tasks that either did, or did not, require overcoming the curse of knowledge, revealing that when the curse of knowledge was removed three-year-olds were significantly better at inferring false beliefs, and as accurate as five- and six-year-olds. These findings reveal that the classic task is not specifically measuring false belief understanding. Instead, previously observed developmental changes in children's performance could be attributed to the ability to overcome the curse of knowledge. Similarly, previously observed relationships between individual differences in false belief reasoning and a variety of social outcomes could instead be the result of individual differences in the ability to overcome the curse of knowledge, highlighting the need to re-evaluate how best to interpret large bodies of research on false belief reasoning and social-emotional functioning.

## Introduction

The capacity to recognize that others have minds, and reason about the contents of those minds, is critical for navigating our social world. Often referred to as a 'Theory of Mind', this

**Funding:** This research was supported by The Natural Sciences and Engineering Research Council grant (2017- 04067), as well as the Social Sciences and Humanities Research Council grant (752-2017- 2358).

**Competing interests:** My coauthors and I have no conflict of interest regarding the manuscript.

capacity is essential for understanding human action [1], it also provides the foundation for cultural learning [2, 3], reduces prejudice [4], and fosters prosocial behavior [5–8]. The ontogeny of this capacity in humans has intrigued philosophers and researchers for decades. When do children understand that others have minds that represent, and therefore can *mis*represent, reality? In some of the earliest work investigating this question researchers used what became known as the Sally-Anne Task (a.k.a., the Classic False Belief task, Maxi Task, or Change-of-Location task; [9]). In this task, children observe a scenario where a protagonist (e.g., Sally) hides an object in one location and leaves the scene. In Sally's absence, another character (e.g., Anne) hides the object in a different location. Then, children are asked where Sally will look for the object when she returns. The logic is as follows: if children can predict that Sally will look for the object where she originally hid it—because she will not know what transpired in her absence and thus will hold a *false* belief—then they understand that a mind can misrepresent reality [9–12]. Across hundreds of replications of this task, and others like it [11–14], 3-year-olds consistently fail by indicating that Sally will first look for the object where it is currently (and where they know it to be), whereas by around 4.5 to 5 years of age, children tend to correctly infer Sally's false belief (see 14, for meta-analysis).

Notably, Theory of Mind is a multifaceted capacity that is much broader than simply passing the Sally-Anne task, or reasoning about false beliefs. A 'theory of mind' is generally believed to involve a suite of abilities like reasoning about emotions, intentions, desires, knowledge, and much more (for review, see [15]). To examine the developmental trajectory of different Theory of Mind abilities researchers have designed various measures such as the Reading the Mind in the Eyes Test [16], Belief-Desire Reasoning task [17], Strange Stories [18], among many others (e.g., [19–23]). Together, these findings reveal that there is important development in Theory of Mind both before and after children succeed at classic false belief tasks.

Despite the existence of several innovations in Theory of Mind measures, the classic Sally-Anne task is still by far the most widely used measure in the literature, especially for preschool-age children. As of 2020, the original work by Wimmer and Perner has been cited 7,733 times [9], and Wellman and Liu's Theory of Mind Scale (which includes a variant of the false belief task) has been cited 1,978 times [24]. For better or worse, the Sally-Anne task remains the gold standard in developmental and cognitive science providing the primary foundation for various fields of research relating to the understanding of the mind such as its neuroanatomical loci [25, 26], its role in Autism Spectrum Disorder [27, 28], and ADHD [see, 29], as well as its various cognitive and environmental correlates [30–34], and its ability to predict a whole host of social outcomes such as one's tendency to exhibit empathy and prosocial behavior [see 8], the degree of one's peer relationship problems [35–37], and one's academic success [38, 39]. Given that performance on this task is such a widely used predictor of various social outcomes, and it forms the bedrock on which several fields of research have been built, it is imperative to understand precisely what it is measuring and which aspects of cognition account for the developmental differences.

Unfortunately, there is still widespread disagreement among researchers about what the Sally-Anne task is measuring, and *why* 3-year-olds fail this classic task [40–42]. One interpretation, sometimes referred to as the 'Fundamental Change' view, is that 3-year-olds do not yet comprehend that others have minds that can misrepresent reality [14, 43, 44]. Another interpretation, sometimes referred to as the 'Processing Demands' view is that 3-year-olds' more general cognitive abilities (e.g., working memory, inhibitory control, language) are not sufficiently developed to allow them to succeed. Under this latter interpretation, it is not that 3-year-olds lack the concept of false beliefs [45–47], but that developmental limitations in cognitive processing abilities undermine their performance. Several variants of this latter interpretation have been put forth emphasizing different cognitive processes but on the whole these

researchers argue that if these processing demands were sufficiently reduced, even very young children could successfully reason about false beliefs (for more accounts, see [48–50]).

Several studies have shown that when the Sally-Anne task is modified in ways that clarify the task, increase the motivation of the participant, or minimize task demands, young children's false belief performance tends to improve. For example, Siegal and Beattie (1991) changed the false belief question to where will the protagonist look for the target object first? As opposed to, where will the protagonist look for the target object. They found that this clarification improved children's false belief performance [51]. Other researchers have shown that when children are more actively involved in the false belief scenario, such as when they are deceiving the protagonist by hiding the target object, young children were more likely to accurately infer the protagonist's false belief [52–56]. Still other researchers have identified that when specific cognitive demands are reduced, including explicit response generation [e.g., 47], executive functioning [57, 58], attentional biases [e.g., 59], and language comprehension [e.g., 60], young children's false belief performance tended to improve [42, 51, 52, 61].

Initially, infant research also appeared to provide support for the view that the Classic Sally-Anne task is unnecessarily cognitively demanding for young children [62–65]. Onishi and Baillargeon were first to demonstrate infants' apparent ability to reason about false belief using an experimental paradigm measuring whether infants looked longer (evidencing surprise) at one of two visual scenes (i.e., one in which an actor behaved in line with a true belief and one with an actor behaving in line with a false belief; [64]). This task placed fewer cognitive demands on participants. For example, the paradigm did not require any understanding of language, it did not require that the participant make a prediction about one's future behavior, nor did it require the participant to intentionally choose a response. In the absence of these demands, even 15-month-olds seemed to understand that people can hold false beliefs [66, 67]. Recently, however, the robustness of these findings has been called into question following several failed replication attempts from multiple labs [68–71]. Moreover, even if the initial findings hold true, it is not the changes *in infancy* that have been shown to predict a host of social outcomes—it is the changes in cognition between three and four years of age that do, and therefore it is those changes that necessitate better understanding.

In the current study, we examine a different cognitive mechanism that has so far received little attention—the effect of the 'curse of knowledge' on children's false belief reasoning. The curse of knowledge is the tendency to be biased by one's own (current) knowledge when reasoning about a more naïve perspective (a.k.a., the 'hindsight bias', 'creeping determinism', 'reality bias', or the 'knew-it-all-along effect'; [72–75]). As a classic example of the curse of knowledge bias, adults who learn the outcome of an event (e.g., an election, a battle, a sports game) overestimate the likelihood that others will predict that outcome. In comparison, adults who were not informed of the event outcome tend to be more accurate in their estimates of what others will predict [73, 76, 77].

In the Sally-Anne task, children are made aware of *exactly where Anne moved the object* even though specific outcome information is not a *necessary* condition to infer a false belief. Accordingly, to pass the Sally-Anne task, children are required to overcome the curse of knowledge and infer Sally's naïve perspective. This is especially difficult for younger children, as they tend to be more vulnerable to the curse of knowledge than older children [78–81]. Still, even adults' false belief reasoning can be impaired by possessing specific outcome information [82]. In their experiment, Birch and Bloom presented participants with a four-container false belief task, where adults were told that a protagonist, named Vicky, was playing her violin and then decided to go outside (Fig 1). Before leaving, she placed her violin inside a blue container. In her absence, participants were told that the locations of the containers were altered, and the violin was moved to another container. Some of the participants were told exactly where the

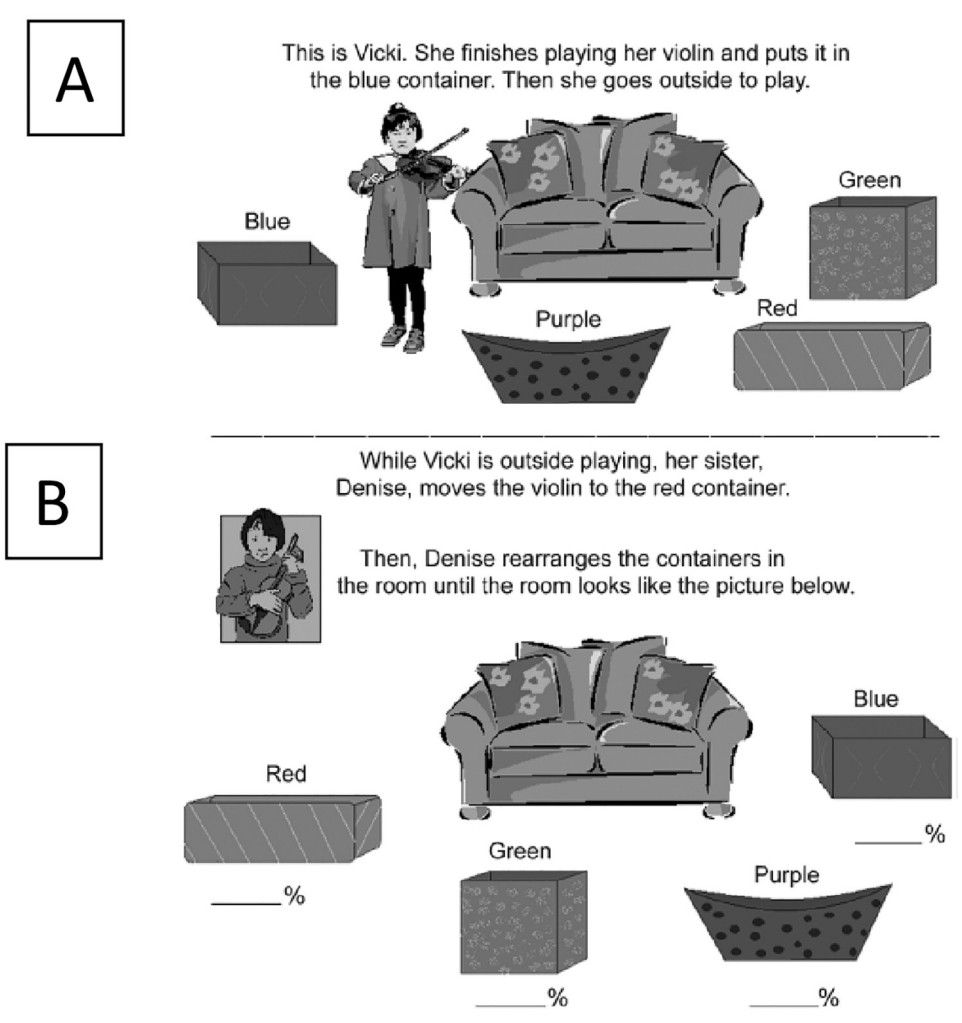

**Fig 1. Stimuli from Birch and Bloom's (2007) study, where adults are given a plausible reason to believe that Vicki would look for her violin in the current container.** Note, that the red container (current container) in section B is in the same spot as the blue container (original container) in section A. From "The Curse of Knowledge in Reasoning About False Beliefs" by Birch & Bloom (2007), *Psychological Science, 18*(5), 384.

violin was moved (e.g., red container), and others were simply told that it was moved to a different container, with no specific information as to where. Participants who knew it had been moved to the red container were significantly worse at predicting Vicky's false belief than participants who did not know where the object had been moved. That is, adults who can unquestionably reason about false beliefs can be biased, or 'cursed' so-to-speak, by their own knowledge when reasoning about false beliefs [for replications and extensions of this work, see 83, 84]. It might be tempting to attribute the curse of knowledge to poor inhibitory control, where an individual is biased by their knowledge *because* they are unable to inhibit the contents of their own knowledge. While there is evidence for the role of inhibitory control on the curse of knowledge [81, 85, 86], inhibitory control does not fully account for the curse of knowledge [87–89]—a point we return to in greater detail in the Discussion.

In the current experiment, we aimed to directly examine the effect of the curse of knowledge on false belief performance by manipulating children's outcome knowledge, while keeping trials consistent with one another. To test this, we employed the logic detailed in Birch (2005) and used by Birch and Bloom (2007) and created two variants of the Sally-Anne task [see Fig 2; 82, 90], where a protagonist hides an object in one of four containers and leaves the scene. As in the standard task, another character moves the object in the protagonist's absence. Our critical manipulation was whether the participants were told specific outcome information: For two test trials (Outcome Known, or Cursed, Trials), we kept it consistent with the standard task and told children exactly where the object was moved (e.g., "to the blue box"). In the other two test trials (Outcome Unknown, or Curse-Lifted, Trials), we told them that the object was moved but did not tell them where (e.g., "to one of the other boxes"). After each test trial, children were asked where the protagonist will look for the object. In other words, the manipulation across test trial types allowed us to test whether eliminating, or reducing, the effect of the curse of knowledge (i.e., in the Outcome Unknown Trials) affected children's false belief reasoning.

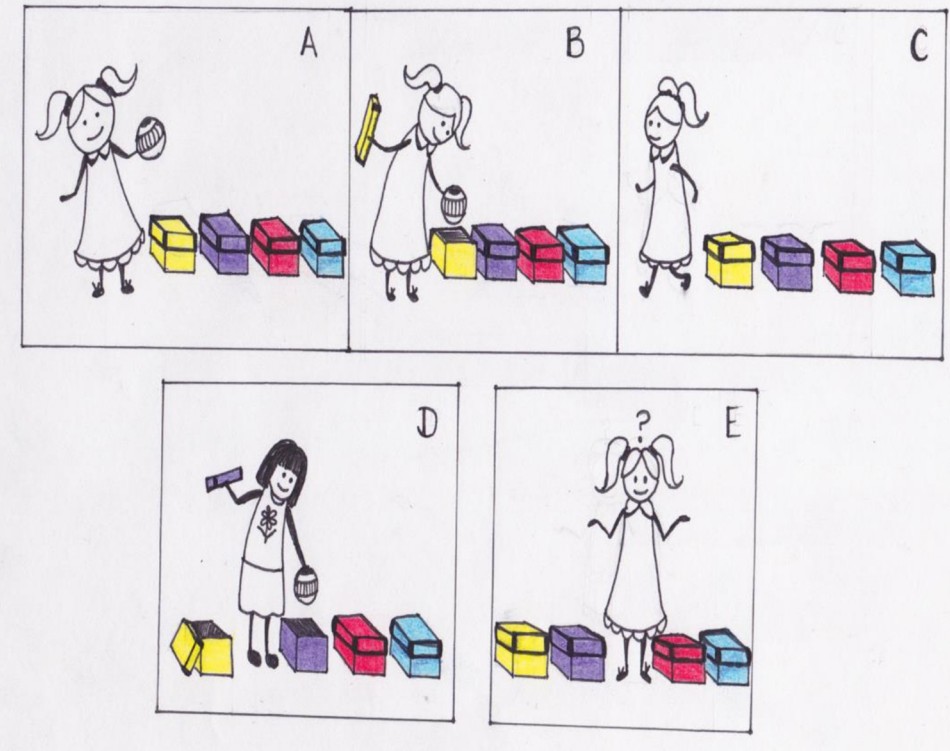

**Fig 2. A depiction of the four-box Sally-Ann task used in the present experiment.** The researcher presented each child with a visual enactment of four false belief scenarios using colorful stick figures. Each scenario involved a protagonist (a) who put an object in one of the four containers (b). Then, the protagonist leaves (c), and another character placed the object in a different container (d). In one scenario for example, the researcher said to the child: "*Sally was playing with her ball, then she got hungry, so Sally put her ball right in here* [purple container], *and went home. When Sally was gone, Ryan hid Sally's ball in a different spot! He may have hid it here, or here, or here*" [researcher pointed to each of the other containers]. In two scenarios (i.e., Outcome Known Trials), the researcher told the child exactly where the second character placed the object (e.g., "*But we know that Ryan hid the ball here.*" [blue container]). In the other two scenarios in contrast (i.e., the Outcome Unknown Trials), the researcher told the child: "*But we don't know where he hid it.*" Both trial types ended the same way (Panel E, *e.g.,* "*Then, Sally came back. Where will she look for her ball?*".

Given previous research suggesting that the curse of knowledge interferes with the ability to reason about false beliefs (e.g., see [76]), we predicted that children would perform more accurately on the false belief task when they are not required to overcome the curse of knowledge. Also, considering that young children are more susceptible to the bias compared to older children and adults [e.g., 79], we hypothesized that the age-related changes in children's performance on the Sally-Anne task may reflect older preschoolers' greater ability to overcome the curse of knowledge bias, rather than a conceptual change in their understanding of the mind.

## Method

### Participants

Once this study was approved by the university's Behavioural Research Ethics Board (ID: H10-01272), 277 three- to six-year-olds were recruited to participate. Thirty-nine of these participants' data were excluded from analyses for reasons that interfered with the validity of their responses. Specifically, 87% percent of the 39 excluded cases were due to a failure to complete the experimental session because they did not answer the questions, explicitly declined to participate, or were extremely inattentive or distracted. An additional 5% of excluded cases involved participants with language comprehension difficulties, 5% involved participants who were interrupted during the experiment, and another 3% had a diagnosis of Autistic Spectrum Disorder. For the remaining two-hundred and thirty-eight cases, we excluded any trial where children could not remember the original location of the object (as will be discussed below and in line with previous false belief research). Thus, 230 children were retained in the final sample ($M_{Age}$ = 54 months, 55% male). Of the 230 children, 37% percent were of Caucasian descent, 17% were of East Asian descent, 12% were of mixed descent (e.g., half East Asian half Caucasian), one child was of Arab descent, and another child was of African American descent. Parents of the remaining 33% of the children did not disclose ethnicity information. Participants were recruited from a multicultural North American city, via visits to community preschools, a local children's museum, and by contacting parents who had previously participated in a university study. Children were considered eligible to participate if they were between 3 and 6 years of age. For analyses, participants were grouped into 3 age-categories to examine potential developmental differences in performance: There were 91 3-year-olds ($M_{Age}$ = 41.5 months, SD = 3.5 months, 56% males), 70 4-year-olds ($M_{Age}$ = 53 months, SD = 3.5 months, 55% male) and 69 5&6-year-olds ($M_{Age}$ = 71 months, SD = 6.5 months, 54% male).

### Materials

The false belief stories were presented to children using stick figures. One laminated paper with an image of four containers (one yellow, one purple, one red, and one blue) was presented during the entire duration of the experiment. A total of eight stick figures were also presented. Each stick figure was a paper cut out of a simple drawing of a person (created using Paint) that was taped on a small stick. Two stick figures (a male and female) were presented for each false belief trial.

### Procedure

Children were tested individually in a quiet area. Each child was presented with four false belief test trials; for each of the test trials, the experimenter presented a scenario about two characters and acted out the scenario with two stick figures (see Fig 2, for a visual illustration of a scenario).

The scenario involved a protagonist who hid an object in one of four containers. Then, in the absence of the protagonist, another character placed the object in a different container. In two of the tasks, children were told exactly where the second character placed the object. For example, the following scenario was presented:

"*Sally was playing with her ball, then she got hungry, so Sally put her ball right in here [**purple** container], and went home. When Sally was gone, Ryan hid Sally's ball in a different spot! He may have hid it here, or here, or here [researcher pointed to each of the other containers]. But we know that Ryan hid the ball here [**blue** container]. Then, Sally came back.*"

In the other two tasks, children were told that the second character moved the object to another container, but they were not told exactly which container. For example, the following scenario was presented:

"*Bill was eating a chocolate bar, but he got full before he could finish it so he put the rest of his chocolate bar in here [**red** container], so he could have it later. Then, Bill went home. When Bill was gone Jane came, and hid his chocolate bar somewhere else! She may have hid it here, or here, or here [researcher pointed to each of the other containers]. But, **we don't know** where she hid it. Then, Bill came back.*"

After each scenario was presented, children were asked where the protagonist will look for the object (e.g. "*Where will Bill look for the chocolate bar?*"), to examine their ability to infer the protagonist's false belief. Then, children were administered two memory checks to ensure they had paid attention to each story: One question assessed whether children could remember where the second character moved the object (e.g., "*Do you know where the chocolate bar is really?*") and another question assessed whether children recalled that the protagonist was absent when the object was moved by the second character (e.g., "*Did Bill see Jane moving the chocolate bar?*"). The majority of participants answered the memory questions correctly. Importantly, children were also asked where the protagonist (e.g., Bill) left the object in the beginning (e.g., "*Do you remember where Bill put the Chocolate bar in the beginning?*"). This question is typically used as an exclusion criterion for Classic False Belief tasks because if the child forgot where the protagonist hid the object in the beginning, then it is impossible to assess their understanding of the protagonist's false belief of the object's location. Accordingly, we only included children's data in our analyses for test trials where they correctly remembered the original location of the object.

In this experiment, we used two versions to alternate which trial type was first presented (i.e., Outcome Known Trial or Outcome Unknown Trial; see Table 1). That is, the same scenarios and the same order of scenarios were used for both versions, however for Version 1 we presented the first scenario as an Outcome Known Trial, and for Version 2 we presented the first Scenario as an Outcome Unknown Trial.

## Results

### Preliminary analyses

Consistent with previous work [14], data used for our main analyses only included performance on test trials where children could recall the original object location for the given trail (e.g., "*Do you remember where Bill put the Chocolate bar in the beginning?*"). Responses to these questions were coded as incorrect (0) and correct (1), resulting in a final sample of 230 three to six-year-old participants (Range = 36 to 83 months; $M_{pass}$ = .72, $SD_{pass}$ = .45).

**Table 1. The false belief scenarios presented by trial type and version.**

| Trial | Version 1 | Version 2 |
|---|---|---|
| 1 | Outcome Known Trial: | Outcome Unknown Trial: |
| | Sally was playing with her ball, then she got hungry, so Sally put her ball right in here (*purple container*), and went home. When Sally was gone, Ryan hid Sally's ball in a different spot! He may have hid it here, or here, or here (researcher *pointed to the other containers*). But we know that Ryan hid the ball here (*blue container*). Then, Sally came back. | Sally was playing with her ball, then she got hungry, so Sally put her ball right in here (*purple container*), and went home. When Sally was gone, Ryan hid Sally's ball in a different spot! He may have hid it here, or here, or here (researcher *pointed to the other containers*). But, we don't know where he hid it. Then, Sally came back. |
| 2 | Outcome Unknown Trial: | Outcome Known Trial: |
| | Bill was eating a chocolate bar, but he got full before he could finish it so he put the rest of his chocolate bar in here (*red container*), so he could have it later. Then, Bill went home. When Bill was gone Jane came, and hid his chocolate bar somewhere else! She may have hid it here, or here, or here (researcher *pointed to the other containers*). But, we don't know where she hid it. Then, Bill came back. | Bill was eating a chocolate bar, but he got full before he could finish it so he put the rest of his chocolate bar in here (*red container*), so he could have it later. Then, Bill went home. When Bill was gone Jane came, and hid his chocolate bar somewhere else! She may have hid it here or here or here (researcher *pointed to the other containers*). But we know that Jane hid the chocolate bar here (*blue container*). Then, Bill came back. |
| 3 | Outcome Known Trial: | Outcome Unknown Trial: |
| | Jenny was reading her book, and then she decided to go to the park. So, Jenny put her book right in here (*yellow container*). When Jenny left, Joe hid the book in a different spot! Joe may have hid it in here, here, or here (researcher *pointed to the other containers*). But we know that Joe hid the book in here (*red container*). Then, Jenny came back to get her book! | Jenny was reading her book, and then she decided to go to the park. So, Jenny put her book right in here (*yellow container*). When Jenny left, Joe hid the book in a different spot! Joe may have hid it in here, or here, or here (researcher *pointed to the other containers*). But, we don't know where Joe hid the book. Then, Jenny came back to get her book! |
| 4 | Outcome Unknown Trial: | Outcome Known Trial: |
| | Rob was playing with his toy car, but he got thirsty and wanted to get some water. Rob put his toy car in here (*blue container*). Then, Rob went home to get some water. Kristy came, and hid the toy car somewhere else! Kristy may have hid the car here, or here, or here (researcher *pointed to the other containers*). But, we don't know where Kristy hid the car. Then, Rob came back. | Rob was playing with his toy car, but he got thirsty and wanted to get some water. Rob put his toy car in here (*blue container*), Then, Rob went home to get some water. Kristy came, and hid the toy car somewhere else! Kristy may have hid the car here, or here or here (researcher *pointed to the other containers*). But we know that Kristy put the car in here (*purple container*). Then, Rob came back. |

Preliminary analyses were conducted on the responses to these memory items to ensure that trial exclusions based on these items were unrelated to our experimental manipulation. We conducted a mixed effect logistic regression with a random intercept for subject to model the repeated measurements in the R environment (see Table 2), where we predicted memory performance from trial type (Outcome Known = 1, Outcome Unknown = 0), gender (0 = females, 1 = males), and age group (Reference group = 3-year-olds, 1 = 4-year-olds, 2 = 5 and 6-year-olds). As can be seen in Table 2, age and gender, significantly predicted memory performance, such that males were less likely to pass the memory trials compared to females, $OR = 0.68$ [0.49–0.96], and older children were more likely to pass the memory trials compared to younger children (4-year-olds, $OR = 1.73$ [1.18–2.55]; 5–6 year-olds, $OR = 3.26$ [2.12–5.02]). Importantly, however, trial type did not significantly predict memory performance. That is, children's ability to pass the memory question did not differ on Outcome Known and Outcome Unknown trials and therefore cannot account for the experimental differences observed below in our main analyses.

**Table 2. Mixed effect logistic regression with a random intercept for subject examining the effect of trial type, gender and age group on memory performance.**

| Predictors | Memory Check (1 = Pass) | | | |
|---|---|---|---|---|
| | Odds Ratios | CI | Statistic | p |
| (Intercept) | 1.99 | 1.41–2.81 | 3.89 | **<0.001** |
| Trial Type (1 = Outcome Unknown) | 1.14 | 0.84–1.54 | 0.85 | 0.396 |
| Gender (1 = Males) | 0.68 | 0.49–0.96 | -2.22 | **0.026** |
| Age group (4-year olds) | 1.73 | 1.18–2.55 | 2.80 | **0.005** |
| Age group (5-6-year olds) | 3.26 | 2.12–5.02 | 5.38 | **<0.001** |
| N $_{Observations}$ | 924 | | | |
| N $_{ID}$ | 230 | | | |

Notes: Reference category for Age group is 3-year-old participants.

## Main analyses

We analyzed children's responses to the false belief questions for each trial (e.g., "Where will Bill look for the chocolate bar?"). We coded their responses as correct (= 1) if they accurately inferred the protagonist's false belief (i.e., indicated that the protagonist will look for the object in its original location), and as incorrect (= 0) if they did not infer the protagonist's false belief (i.e., indicated that the protagonist will look for the object in a different location). We conducted a mixed effect logistic regression with a random intercept for participant ID to account for the repeated measures predicting false belief performance. Fixed effect predictors included were trial type (0 = Outcome Known, 1 = Outcome Unknown), gender (0 = females, 1 = males), age group (Ref group = 3 year olds, 4-year olds, 5-6-year olds), and version (0 = version 1, 1 = version 2; see Table 1 for version differences). Given our prediction that younger children would benefit the most from the reduced curse of knowledge, we also examined the interaction between age group and trial type.

Fig 3 illustrates our focal result (see Table 3 for model summary) that 3-year-olds were predicted to perform equally well as older children (4- and 5-6-year-olds) on the false belief trials when the location was *unknown*–and that when the location was *known* (as is in the classic False Belief task), performance in the younger children was significantly worse. That is, when not cursed by the outcome knowledge, younger children were just as capable of inferring false beliefs as older children. As expected given previous literature [30, 91, 92], we found that males performed on average worse than females across the board (see Table 3). Crucially, an additional analysis indicated that when children failed the Outcome Known Trials, they tended to choose the *current* location of the object 80% of the time, which is significantly above chance, t(138) = 20.44, p < .001, providing evidence that participants failed the Outcome Known trials *because* they were biased by their knowledge of exactly where the object was moved (i.e., they were not responding randomly).

Interestingly, 5- and 6-year-olds who typically perform fairly well on the classic 2-container versions of the Sally-Anne task (i.e., they pass at around a 65% average and a 75% average, respectively; [14]) were passing at a lower rate on the current 4-container versions for both our Outcome Known and Outcome Unknown trials. Of course, unlike a 2-container task the current task cannot be passed by chance 50% of the time, but only 25% of the time. In other words, in previous research children had a greater probability of succeeding by chance without ever reasoning about a false belief. That is not to say that older children do not understand false beliefs; they clearly do (i.e., they perform significantly above chance), but simply that performance on a 2-box task is less likely to capture children's difficulties than a task where the

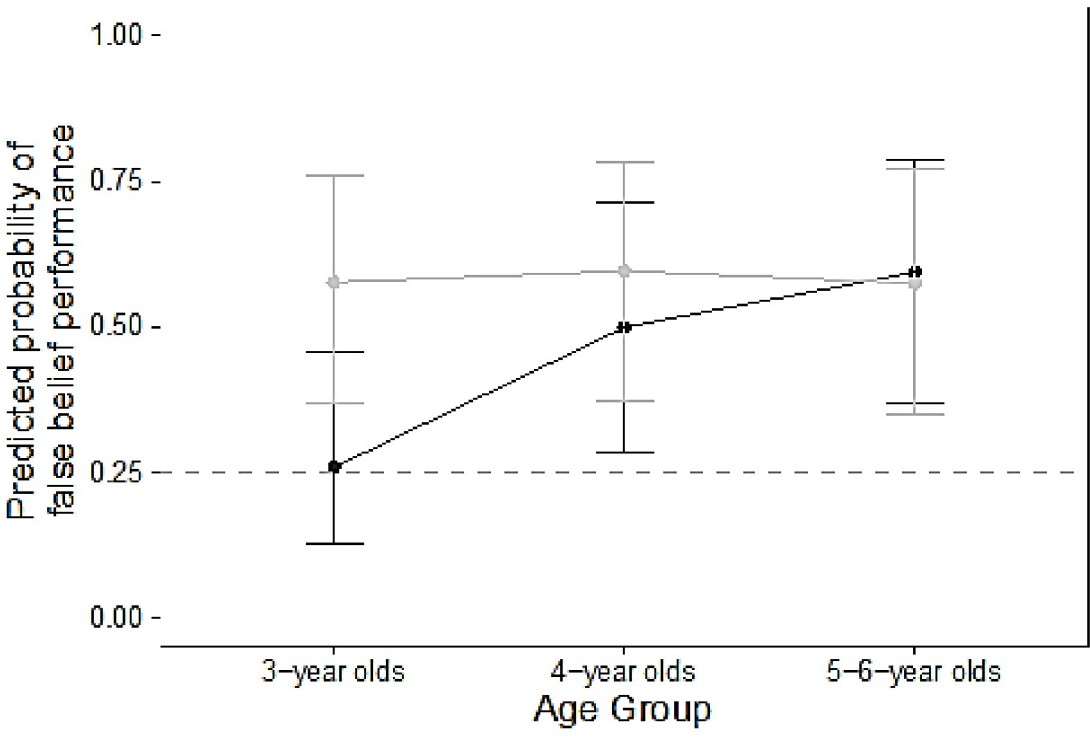

**Fig 3. The predicted probabilities of false belief performance by age group and trial type (predictions from model presented in Table 3).** 3-year-olds' performance was significantly less accurate in the Outcome Known trials (i.e., when specific outcome information was presented; estimated.26 probability of passing each trial) versus Outcome Unknown Trials (i.e., when specific outcome information was *not* presented; estimated.58 probability of passing each trial). In the older age groups, predicted probability of passing did not significantly differ by trial type (predicted probabilities ranging from.50 to.60). Notably, when the outcome was unknown young children performed equally as well as older children. Error bars are 95% confidence intervals around the predictions, and the dashed line represents chance performance given that there were four containers. Note that all age groups were similarly predicted to perform above chance on Outcome Unknown trials.

**Table 3. Mixed effect logistic regression with a random intercept for participant predicting false belief performance.**

| Predictors | False Belief (1 = Pass) | | | |
| --- | --- | --- | --- | --- |
| | *Odds Ratios* | *CI* | *Statistic* | *p* |
| (Intercept) | 0.34 | 0.14–0.82 | -2.40 | **0.017** |
| Trial Type (1 = Outcome Unknown) | 3.91 | 1.64–9.31 | 3.08 | **0.002** |
| Age group [4-year olds] | 2.86 | 0.80–10.20 | 1.62 | 0.105 |
| Age group [5-6-year olds] | 4.21 | 1.18–15.03 | 2.21 | **0.027** |
| Sex (1 = Males) | 0.40 | 0.16–0.96 | -2.05 | **0.041** |
| Version (1 = Version 2) | 0.59 | 0.24–1.48 | -1.12 | 0.264 |
| Trial Type * 4-year olds | 0.38 | 0.11–1.25 | -1.59 | 0.112 |
| Trial Type * 5-6-year | 0.24 | 0.07–0.77 | -2.39 | **0.017** |
| N Observations | 667 | | | |
| N ID | 230 | | | |

probability of passing by chance is lower. Importantly, while 5-and 6-year-olds tend to do much *better* at reasoning about false beliefs than younger children, they are by no means perfect [14]. Poor false belief performance in older children on modified Sally-Anne tasks is consistent with other work [50, 93]. For example, Fabricius and Khalil (2003) found that adding an additional container to the Sally-Anne task significantly worsens 4 to 6-year-old children's false belief performance. Specifically, in one of their experiments, the researchers found that eighty-four 4.5 to 5.5-year-old children were 71% accurate when responding to the classic 2-box Sally-Anne task (i.e. 21% better than chance), however they were only 55% accurate when responding to a modified 3-container false belief task (i.e. 22% better than chance). Comparatively, our older children performed on par, or better, in our 4-box task (i.e. 34% better than chance) as the older children in the 3-box task. The point here is simply to remind the reader that even older children are not perfect at false belief reasoning. Importantly, however, removing the curse of knowledge from false belief tasks allows 3-year-olds to perform on par with 5-and-6-year-olds.

## Discussion

Our results demonstrate that 3-year-old children are significantly more likely to accurately infer a false belief when they do not have specific outcome information, compared to when they have specific outcome information. When young children knew the true location of the object, they tended to indicate that the protagonist will look for the object in *that* location; that is, they tended to fail the task because they were biased or 'cursed' by their own knowledge. Age analyses revealed that the curse of knowledge only affected young children's false belief reasoning abilities, and not those of older children. Critically, these results illustrate that when the curse of knowledge is minimized from these measures by removing specific outcome information, *3-year-olds are equally as good as older preschoolers at reasoning about others' false beliefs*.

Importantly, our main finding showing that younger children infer false beliefs more accurately, when they are not cursed by their knowledge, echoes a growing body of literature suggesting that the Classic Sally-Anne task *underestimates* children's ability to reason about the mind. We are not the first to make this latter claim, leading some to suggest the task should be abandoned [e.g., 45]. However, instead of abandoning the task, we advocate for a) revisions to the task to minimize the curse of knowledge where the intent is to specifically measure false belief reasoning and b) reinterpretations of the substantive bodies of literature that have relied exclusively on the standard ('cursed') Sally-Anne task [e.g., see 94]. Abandoning the task would be a mistake because performance on the task has proven to be a reliable predictor of an abundance of critical social outcomes [e.g., 37, 95, 96]. But as our results reveal, the Sally-Anne task is not only measuring false belief reasoning, but it is also children's ability to overcome the curse of knowledge. For instance, the difference in children's performance on the standard task at age 3 versus age 5 appears to stem more from developmental change in the magnitude of the curse of knowledge rather than a developmental change in children's understanding of false beliefs. Similarly, the various predictors of false belief performance that researchers have found (e.g., presence of siblings, exposure to mental state discourse) may more accurately be predictors of the magnitude of the curse of knowledge [97–99]. The same argument applies to research examining the relationships between individual differences in false belief performance and various social-emotional outcomes and indices (e.g. bullying, communication, prosocial behavior, peer relationships, social-emotional well-being; e.g., [95, 96]). We raise the possibility that the abundance of literature that has relied on false belief performance as a predictor of these social outcomes is tapping into differences in one's ability to overcome the curse of knowledge, rather than false belief reasoning per se.

False belief reasoning and the curse of knowledge are not completely unrelated constructs —they both involve social perspective taking. The curse of knowledge is far more pervasive in social interactions than one's ability to reason about false beliefs specifically—it is present any-time you need to reason about a more naïve perspective. The detrimental impact of the curse of knowledge bias has been documented in many applied settings including business, educa-tion, politics, communication (both oral and written; and in medical, legal, and government decision-making). The similarity between the two constructs may have confounded previous research examining the Sally-Anne task, as this research may have been inadvertently measur-ing variance in the ability to overcome the curse of knowledge rather than the ability to infer a false belief. The similarity between these two constructs also serves to highlight why young children make more mistakes than older children and adults in everyday situations that involve perspective taking: Young children *are* more apt to make perspective-taking errors— not because they lack a concept of false beliefs, but because they are more cursed by their knowledge.

Importantly, previous research has hinted at the effect of the curse of knowledge on young children's false belief reasoning. For example, to reduce inhibitory control demands, Setoh, Scott and Baillargeon presented children with a variant of the Sally-Anne task, where children were not told the current location of the target object, and therefore children did not have to inhibit a prepotent response of pointing to the actual location of the target object [42, 100]. The researchers also included two practice questions to help children practice generating responses. In this task, the researchers told children about Emma who found an apple in a bowl, and she moved it to a box. Then, the researchers presented children with two practice questions, each question involved showing children two pictures of two different items (e.g., an apple and a banana), and asking them to point at one of the items (e.g., 'where is Emma's apple?'). Then, the researchers told children that in Emma's absence, Ethan came, and he took away the apple, and then they asked children where Emma will look for her apple.

Using this task, Setoh, Scott and Baillargeon (2016) found that 2.5-year-olds can pass the task and indicate that Emma will look for her apple in the bowl. However, when children were told the current location of the apple, they pointed to the current location significantly above chance. The researchers interpreted this finding as indicative of an inhibitory control account, where young children have not fully developed the cognitive ability to inhibit the prepotent response to point at the current location. These findings can also be interpreted under the curse of knowledge framework, where children have not yet developed the cognitive resources to realize that Emma does not share their knowledge, and she does not know that her apple is in the current location.

Other researchers have created a reality-unknown variant of the task [101] or reduced the saliency of the 'current location' (the one that ultimately 'curses' the child) by telling children that the secondary character ate the target object (here the target object is chocolate; [54]). These studies have led to some mixed findings. Koos and colleagues (1997) reported an improvement in 3-year-olds' performance on the 'saliency-reduced' version compared to the Sally-Anne task, whereas Call and Tomasello (1999) found that their reality-unknown task is difficult for even 4-year-olds, and is as difficult as the Sally-Anne task. While these studies offer novel and important ways of examining false belief reasoning, they do not systematically examine the effect of outcome knowledge on children's false belief reasoning. One study remains unpublished [54], and the other [101] utilized a paradigm that is very different from the Sally-Anne task and is difficult for other reasons [58, 101].

To our knowledge, our study is the first to directly examine the effect of the curse of knowl-edge on false belief reasoning among children using identical tasks that vary solely based on whether outcome information is presented or not. As such we contribute a new measure that

will allow researchers to specifically examine false belief reasoning rather than the ability to overcome the curse of knowledge. Importantly, this measure can capture younger children's ability to reason about false beliefs, as well as older children. In addition, using both versions of the false belief task (Outcome Known vs Outcome Unknown) will allow researchers to specifically address whether, and which, children struggle with overcoming the curse of knowledge. We advocate for the use of this 'curse-lifted' variant, in research examining the relationship between children's social perspective taking and social outcomes, instead of relying solely on the Sally-Anne task which we argue is 'cursed', so-to-speak, because it does not specifically tap into false belief reasoning.

In understanding the role of the curse of knowledge in perspective taking and the nature of the cognitive changes occurring between ages three and four, it is important to consider the mechanisms that contribute to the curse of knowledge bias. As mentioned in the introduction it is intuitive to imagine that the curse of knowledge is the result of deficits in inhibitory control. Although some researchers have advocated for this account [e.g., 96], inhibitory deficits alone are insufficient to account for much of the data demonstrating the conditions under which the bias does, and importantly *does not*, occur. In our view, a fluency misattribution account is necessary in understanding the curse of knowledge. A fluency misattribution account suggests that when information is processed fluently (i.e., it is easy to process because it has been encountered before or it makes sense), we tend to mistake this ease in processing as indicative of how easy or obvious this information is to others [87, 88, 102].

As an example, consider the Birch and Bloom (2007) adult false belief study mentioned above. Recall that some adults were told that Vicky's violin was moved to the red container (Fig 1B; [82]), which occupied the location of the original container where she left it, and therefore it is plausible that Vicky would look for her violin in that location (Knowledge-Plausible Condition). However, in another condition within the same study, the Knowledge-Implausible Condition (not described above), participants were told that the violin was moved to a different container (i.e., the purple container, Fig 1) that did not occupy the old location of the original container. So, in the Knowledgeable-Implausible Condition, participants did not have a plausible reason to believe that Vicky would look for her violin in that container. Only participants in the Knowledge-Plausible Condition showed the curse of knowledge in their false belief reasoning. That is, only participants in the Knowledge-Plausible Condition, and not participants in the Knowledge-Implausible Condition, overestimated the likelihood that Vicky will look for her violin in its current container. Accordingly, learning new information is not enough to lead to the curse of knowledge, the new information must make sense, or it must fit fluently with what an individual already knows.

Similarly, children do not show a curse of knowledge anytime they learn new information, the new information must be familiar for the bias to occur [89]. In two experiments, researchers asked children to estimate how many of their peers would know the answers to various factual questions. Importantly, some of the children learned familiar answers to factual questions, and others learned unfamiliar answers to the same factual questions. Children were only biased in their estimates of what their peers would know, when they were taught familiar answers to the factual questions. This finding, along with many others [e.g., 82, 88, 98], highlight the role of fluency misattribution in the curse of knowledge. While it is possible that inhibitory control may moderate the curse of knowledge [see 81, 85, 103], the bias appears dependent on the fluency of the information acquired.

Fluency *misattribution* stems from an otherwise useful metacognitive process, known as fluency *attribution*. Individuals tend to use the fluency of information to guide memory inferences about familiarity, or previous exposure, to a stimulus [e.g., 104–106]. In other words, when we process a piece of information fluently, it is reasonable to infer that we have been

exposed to this information in the past. Fluency *misattribution* occurs when we associate the ease of certain information to an inaccurate source (e.g., when we mistakenly infer that the ease of the information is attributed to how objectively obvious it is). The role of fluency misattribution on children's social perspective taking has been largely neglected thus far in the developmental literature on social cognition but will be an especially fruitful avenue for future research.

In sum, we argue that research examining the development of false belief reasoning and its predictive power will benefit from a) utilizing new measures that do not require children to also overcome the curse of knowledge bias when inferring a false belief (such as the Outcome Unknown False Belief task used here) and b) reinterpreting previous literature bearing in mind that what was previously considered the result of differences in the ability to reason about false beliefs, may be the result of differences in the ability to overcome the curse of knowledge.

## Supporting information

**S1 File.**
(TXT)

**S1 Raw data.**
(CSV)

## Author Contributions

**Conceptualization:** Susan A. J. Birch.

**Data curation:** Siba Ghrear.

**Formal analysis:** Adam Baimel.

**Funding acquisition:** Susan A. J. Birch.

**Investigation:** Siba Ghrear, Taeh Haddock, Susan A. J. Birch.

**Methodology:** Siba Ghrear, Susan A. J. Birch.

**Project administration:** Siba Ghrear.

**Supervision:** Susan A. J. Birch.

**Visualization:** Siba Ghrear, Adam Baimel.

**Writing – original draft:** Siba Ghrear.

**Writing – review & editing:** Siba Ghrear, Adam Baimel, Susan A. J. Birch.

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
