## [Decision Letter · Decision Letter 0]

7 Oct 2020

PONE-D-20-21529

Are the Classic False Belief Tasks Cursed? Young children are just as likely as older children to pass a false belief task when they are not required to overcome the Curse of Knowledge

PLOS ONE

Dear Dr. Ghrear,

Thank you for submitting your manuscript to PLOS ONE. After careful consideration, we feel that it has merit but does not fully meet PLOS ONE’s publication criteria as it currently stands. Therefore, we invite you to submit a revised version of the manuscript that addresses the points raised during the review process.

We look forward to receiving your revised manuscript.

Kind regards,

Zhidan Wang, Ph.D

Academic Editor

PLOS ONE

Journal Requirements:

2. Thank you for including your ethics statement: '- UBC Behavioural Research Ethics Board

- H10-01272

- Parents' written and oral consent, as well as the child participants' oral consent'

a. Please amend your current ethics statement to include the full name of the ethics committee/institutional review board(s) that approved your specific study and confirm that your named institutional review board or ethics committee specifically approved this study.

3.Thank you for stating the following in your Competing Interests section: 

[Not Applicable].

Reviewers' comments:

Reviewer's Responses to Questions

**Comments to the Author**

1. Is the manuscript technically sound, and do the data support the conclusions?

Reviewer #1: Yes

2. Has the statistical analysis been performed appropriately and rigorously? 

Reviewer #1: No

3. Have the authors made all data underlying the findings in their manuscript fully available?

Reviewer #1: Yes

4. Is the manuscript presented in an intelligible fashion and written in standard English?

Reviewer #1: Yes

5. Review Comments to the Author

Reviewer #1: assessing children's understanding of false beliefs has been a major focus of developmental psychology for more than 30 years. There have been many attempts to demonstrate young preschoolers to be able to show false belief understanding but the standard method somehow hinders their performance. The present study is one of such attempts. I believe the present study makes a useful contribution to the literature by illustrating that the curse of knowledge is one of the major reasons that prevent some children from passing the classic false belief tasks. However, I would like to make the following suggestions that the authors may consider to revise their paper.

First, I believe the authors need to cite some of the earlier attempts to illustrate 3- and 4-year-olds are able to pass the classic tasks if the tasks are revised somewhat. For example, the work by Siegal.

Second, the reporting of the logistic analysis results is not informative. I think the authors need to provide first the statistical results of each block of each model, and then report the beta, and wald chisquare and then odds ratio (the authors only reported the p value and odds ratio) for each of the predictor variables.

Third, The authors noted that the performance of the older children did not reach ceiling. They also correctly noted that most children above 5 years of age in North America would pass the classic false belief tasks. However, I am not convinced that this difference was due to the fact that the chance level of performance was lowered to 25% instead of 50%. To be convincing, the authors need to demonstrate that when the chance level is set the same for the new and the classic tasks (50% or 25%), the removal of the curse of knowledge leads the same level of performance. In fact, I would be more convinced that when the chance levels are the same, the removal of the curse of knowledge would still lead to improvement of performance.

6. PLOS authors have the option to publish the peer review history of their article (what does this mean?). If published, this will include your full peer review and any attached files.

Reviewer #1: No

---

## [Author Response · Author response to Decision Letter 0]

26 Nov 2020

Dear Dr. Wang,

We would like to begin by thanking you very much for the opportunity to revise and resubmit to PLOS ONE. We would also like to thank our reviewers for their time, attention, and suggestions. We carefully considered each of the comments and responded to each below. First, we report the comment, and then our response.

In addition to the changes noted in our response to reviewers, we made a couple of other minor changes to make our manuscript clearer and more succinct (e.g. avoid redundancy). Specifically, we removed what was Table 3 and lines 397-410 in the original submission of our manuscript— that information is now included in the participants section or presented in Figure 3 or the Figure 3 caption.

In the process of our revisions, we noted and corrected, an error involving 8 (of our 277) participants on the memory exclusion criteria that was used to confirm children were attending to, and remembering the critical manipulation, see lines 334-346, under Procedure. This correction did not have any impact on the results, but we mention it here so you know why the numbers have changed slightly in the updated analyses.

Thank you again for your time and attention. 

Sincerely,

Authors

Reviewer 1: First, I believe the authors need to cite some of the earlier attempts to illustrate 3- and 4-year-olds are able to pass the classic tasks if the tasks are revised somewhat. For example, the work by Siegal.

Response: Thank you for this suggestion; that’s an important addition. We added the paragraph below on page 5, where we discussed several examples and provided readers with citations for reviews and metanalyses on this body of work:

“Several studies have shown that when the Sally-Anne task is modified in ways that clarify the task, increase the motivation of the participant, or minimize task demands, young children’s false belief performance tends to improve. For example, Siegal and Beattie (1991) changed the false belief question to where will the protagonist look for the target object first? As opposed to, where will the protagonist look for the target object. They found that this clarification improved children’s false belief performance (59). Other researchers have shown that when children are more actively involved in the false belief scenario, such as when they are deceiving the protagonist by hiding the target object, young children were more likely to accurately infer the protagonist’s false belief (60–64). Still other researchers have identified that when specific cognitive demands are reduced, including explicit response generation (e.g., 55), executive functioning (65,66), attentional biases (e.g., 67), and language comprehension (e.g., 68), young children’s false belief performance tended to improve (50,59,60,69).”

Reviewer 1: the reporting of the logistic analysis results is not informative. I think the authors need to provide first the statistical results of each block of each model, and then report the beta, and wald chisquare and then odds ratio (the authors only reported the p value and odds ratio) for each of the predictor variables.

Response: Thanks for the feedback. To clarify, for our preliminary (memory check) and focal analyses (false belief performance), we conducted a total of two mixed effect logistic regressions that each included a random intercept on participant ID to appropriately account for the repeated measurements within individuals. The tables have been updated to include the odds ratios, the confidence intervals associated with these ratios, the test statistic and the p-value for each predictor. With regards to our focal results, we focus on interpreting the predicted probabilities of passing the false belief test rather than on interpreting the model coefficients themselves, and have succinctly summarised our focal findings in Figure 7. This is in following with best practices for interpreting the predictions of a logistic regression model (McElreath, 2020).

We removed the region of significance tests Figure which was conducted on age months since the primary analyses examined age as a categorical rather continuous variable (i.e. to test the hypotheses that three-year-olds would be as good as older children when the curse of knowledge was removed). It seemed redundant to report two sets of analyses by treating age categorically in one analysis and continuously in another analyses, however, we are happy to report the analyses with age as a continuous variable in a supplementary section if you feel there is added value in its inclusion.

Reviewer 1: Third, the authors noted that the performance of the older children did not reach ceiling. They also correctly noted that most children above 5 years of age in North America would pass the classic false belief tasks. However, I am not convinced that this difference was due to the fact that the chance level of performance was lowered to 25% instead of 50%. To be convincing, the authors need to demonstrate that when the chance level is set the same for the new and the classic tasks (50% or 25%), the removal of the curse of knowledge leads the same level of performance. In fact, I would be more convinced that when the chance levels are the same, the removal of the curse of knowledge would still lead to improvement of performance.

Response: If we understand the reviewers’ concern correctly then we have not been clear enough in either a) the description of our analyses or b) our discussion of one possible explanation for the older children’s relatively poorer performance compared to earlier working using the 2-box classic tasks (or both). We have attempted to clarify each as follows:

a) Specifically, The reviewer wrote: “To be convincing, the authors need to demonstrate that when the chance level is set the same for the new and the classic tasks (50% or 25%), the removal of the curse of knowledge leads the same level of performance.” We wish to clarify that the chance level here is the same in our analyses for both the new (no curse) version of our task and the cursed version of the task (the one most similar to the classic task): 25% since there are 4 response options/4 boxes). In our analyses, the removal of the curse of knowledge leads to the same level of performance in 3-year-olds as in 5-year-olds (see Figure 7). That is, both of our tasks include 4 possible response options (i.e. 4 boxes instead of the classic 2 box task) in order to a) create the possibility of not knowing where the object was moved by process of elimination and b) make the two tasks and analyses comparable as the reviewers suggest. Given the 4 response options for both tasks it would not be statistically correct to set chance at 50% in our analyses.

b) We suspect what may have led to confusion was our discussion of our 5-year-olds performance, and the fact that previous results typically report that 5-year-olds understand false beliefs. Importantly, however, ‘understanding’ (i.e. doing better than chance) is not the same as being good or even great. We are also not saying that older children only pass classic false belief tasks because chance is 50%. To clarify these points, we have added to pages 19-20:

“Interestingly, 5- and 6-year-olds who typically perform fairly well on the classic 2-container versions of the Sally-Anne task (i.e., they pass at around a 65% average and a 75% average, respectively; 14) were passing at a lower rate on the current 4-container versions for both our Outcome Known and Outcome Unknown trials. Of course, unlike a 2-container task the current task cannot be passed by chance 50% of the time, but only 25% of the time. In other words, in previous research children had a greater probability of succeeding by chance without ever reasoning about a false belief. That is not to say that older children do not understand false beliefs; they clearly do (i.e. they perform significantly above chance), but simply that performance on a 2-box task is less likely to capture children’s difficulties than a task where the probability of passing by chance is lower. Importantly, while 5-and 6-year-olds tend to do much better at reasoning about false beliefs than younger children, they are by no means perfect (14). Poor false belief performance in older children on modified Sally-Anne tasks is consistent with other work (58,100). For example, Fabricius and Khalil (2003) found that adding an additional container to the Sally-Anne task significantly worsens 4 to 6-year-old children’s false belief performance. Specifically, in one of their experiments, the researchers found that eighty-four 4.5 to 5.5-year-old children were 71% accurate when responding to the classic 2-box Sally-Anne task (i.e. 21% better than chance), however they were only 55% accurate when responding to a modified 3-container false belief task (i.e. 22% better than chance). Comparatively, our older children performed on par, or better, in our 4-box task (i.e. 34% better than chance) as the older children in the 3-box task. The point here is simply to remind the reader that even older children are not perfect at false belief reasoning. Importantly, however, removing the curse of knowledge from false belief tasks allows 3-year-olds to perform on par with 5-and-6-year-olds.”

---

## [Editor Report · Decision Letter 1]

4 Dec 2020

Are the Classic False Belief Tasks Cursed? Young children are just as likely as older children to pass a false belief task when they are not required to overcome the Curse of Knowledge

PONE-D-20-21529R1

Dear Dr. Ghrear,

We’re pleased to inform you that your manuscript has been judged scientifically suitable for publication and will be formally accepted for publication once it meets all outstanding technical requirements.

Kind regards,

Zhidan Wang, Ph.D

Academic Editor

PLOS ONE
---

## [Editor Report · Acceptance letter]

8 Jan 2021

PONE-D-20-21529R1 

Are the Classic False Belief Tasks Cursed? Young children are just as likely as older children to pass a false belief task when they are not required to overcome the Curse of Knowledge. 

Dear Dr. Ghrear:

I'm pleased to inform you that your manuscript has been deemed suitable for publication in PLOS ONE. Congratulations! Your manuscript is now with our production department. 

Kind regards, 

on behalf of

Dr. Zhidan Wang 

Academic Editor

PLOS ONE